# Racism in public health authorities–A scoping review and situational analysis

Lisa Wandschneider [ORCID][1,2], Sigsten Stieglitz [ORCID][1,3,4], Theresa Altmiks[1], Oliver Razum [ORCID][1,5]*, Julia Zielke [ORCID][1,6], Yudit Namer [ORCID][1,7]

**1** Department of Epidemiology and International Public Health, School of Public Health, Bielefeld University, Bielefeld, Germany, **2** Department of Health Monitoring and Prevention, Public Health Department of Lower Saxony, Hanover, Germany, **3** Department of Medical Psychology, Amsterdam University Medical Center, Amsterdam, the Netherlands, **4** Center of Expertise on Gender Dysphoria, Amsterdam University Medical Center, Amsterdam, the Netherlands, **5** Research Institute Social Cohesion, Bielefeld, Germany, **6** Einstein Center Population Diversity, Charité – Universitätsmedizin Berlin, Institute for Medical Sociology and Rehabilitation Science, Berlin, Germany, **7** Department of Psychology, Health and Technology, University of Twente, Enschede, the Netherlands

* oliver.razum@uni-bielefeld.de

## Abstract

Racism is a structural determinant of health. While racism in health care services is increasingly well-researched, public health services and public health authorities (PHA) have been neglected as institutional contexts. Yet, PHA play an essential role in protecting and promoting population health at a local and national level. To help fill this research gap, we mapped the academic discourse on racism in PHA with a narrative scoping review. We searched in PubMed, Embase, PsycINFO and CINAHL as well as the reference lists of retrieved publications. We included literature assessing racism in interactions between all stakeholders involved or in the actions (not) taken by PHA, while we excluded literature exclusively focusing on health care facilities. We applied situational analysis (SA) for interpreting the scientific discourse on racism in PHA. Our search yielded 55 publications that include survey and intervention studies as well as opinion pieces. Most of the literature focuses on the US, Australia and New Zealand/Aotearoa. The Tuskegee experiment has been discussed extensively acknowledging the political and historical elements of the racist, inhumane practices and policies in PHA. More recent literature explores anti-racism approaches and how they can facilitate access for racialized, socially multiply stigmatized groups (e.g., screening uptake in racialized queer people). SA also suggests racialized groups may be implicated or silenced groups in racism research surrounding PHA. We show that the literature on racism in PHA is limited, mostly processing historical policies. Studies on how racism affects equitable access to PHA and the associated health inequities are lacking. Positive examples highlight the importance of a) building the services in PHA on anti-racism and equity-driven principles and b) integrating and amplifying the voices of racialized community. Public health research on racism

**Data availability statement:** All data collected are from public data bases and freely available.

**Funding:** This work has been completed within the Institutions & Racism Consortium, which is funded by the Bundesministerium des Inneren, für Bau und Heimat (Federal Ministry of the Interior, for Construction and Community), Germany. OR and YN are the grant recipients. There was no grant number. The funders had no role in study design, data collection and analysis, decision to publish, or preparation of the manuscript. We acknowledge support for the publication costs by the Open Access Publication Fund of Bielefeld University and the Deutsche Forschungsgemeinschaft (DFG). Neither the Open Access Publication Fund of Bielefeld University nor the Deutsche Forschungsgemeinschaft (DFG) had any role in study design, data collection and analysis, decision to publish, or preparation of the manuscript.

**Competing interests:** The authors have declared that no competing interests exist.

needs to extend the scope from health care services to the under researched public health services and authorities.

## Introduction

### Racism as a structural determinant of health

Racism is a structural determinant of health. It goes beyond the notion of race as a biological category, which is outdated [1]. We here use the term racialization which refers to the process of societal attribution of racial meanings. This process is not only interpersonal; it often becomes institutionalized and appears permanently ingrained in societal structures. Closely tied to racialization is stigmatization, which manifests in the marginalization, exclusion, and punitive treatment of racialized minority groups. Racism – when defined as a hegemonic structure - perpetuates these processes, reinforcing group distinctions and entrenched hierarchies, which in turn drive health inequities—even in the absence of overt individual racial biases [2,3]. Different forms of racialization, e.g., Anti-Black racism, antisemitism, anti-Muslim and anti-Roma, can shift historically and depend on the context as well as the (in-)visibility of 'other' racialized groups [4]. Racialization and racism are thus embedded in societies and cannot be isolated from additional intersecting systems of oppression, e.g., heteronormativity, classism and sexism [5].

"Racism kills" – is the conclusion of a steadily growing body of evidence [6–9]. Health inequalities research identified significant correlation between belonging in a racialized group and outcomes such as premature death as well as prevalence of diseases like cancer or cardiovascular disease [10,11]. Experiences of racism can lead to chronic stress, anxiety and depression which can have a negative impact on physical health, and mental well-being of millions of people, preventing them from attaining their highest level of health [6].

The growing awareness of structural racism has led to calls for public health researchers, governmental public health practitioners, medical care providers, and policy makers to explicitly identify structural racism as a root cause of racial health inequity [11–13]. This is also reflected in increasing efforts to operationalize and measure racism in public health research [2,14–17]. This research should also extend to structural and institutional racism in health services. Racism in health care settings is increasingly well-researched indicating that first and foremost, patients (as well as health professionals) experience racism in the health care setting. Secondly racism is associated with health care use, e.g., delayed/not getting healthcare, lack of adherence to treatment and reduced participation in prevention [10], adverse birth outcomes [18] and limited access to reproductive health care [19–21].

### Public health authorities – A blind spot in public health research on racism?

Racism in public health authorities (PHA) seems to be under-researched [22]. Depending on the context, different terminology is used to describe PHA, including public health agency, public health department and public health office. Both

within and across countries, states and communities, the scope and scale of services and organizational structures vary widely. In a general sense though, PHA can be described as a government body tasked with protecting and promoting the populations' health. This includes for example the regulation of infectious diseases, sex work, access to abortion, shared accommodation for refugees, as well as social psychiatry. These services are often stigmatized and intertwined with racialization [23,24], since for example only specific groups are tested for infectious diseases (e.g., refugees are tested for tuberculosis but not citizens arriving from the same destinations [25].

PHA and their service providers not only act on interpersonal levels but also mobilize community resources and advocate for policy and law changes to improve community health. Their legal powers play an important role in assuring equitable health access and protection. For example, they regulate quarantine at home (in the case of COVID-19), grant access to abortion, or give permission to engage in sex work. Such powers can be misused when applied in inequitable or unjust ways [26,27]. Encounters with PHA and their service providers can carry significant stigma and disproportionally affect already stigmatized and multiply marginalized groups [26,27]. PHA therefore have a lot of responsibility and power in terms of deciding on the trade-offs between a reduction of individual freedom and autonomy on the one hand and (short-term) protection of population health on the other hand.

Furthermore, PHA are responsible for population health surveillance. Data is needed to identify health inequalities, but it is often categorized in racialized ways. For example, data on the distribution of infectious diseases in different communities become racialized when it classifies certain communities as disease bearers, and as putting others at risk, e.g., of tuberculosis or HIV [28]. Instead of helping ethnic minority communities with surveillance, the collection of data can lead to the perpetuation of racialized discourses.

## Challenging the epistemic silencing of racialized experiences and practices in PHA

Considering the increasingly well documented effects of racism on health and experiences in health care settings [10,29], we aim to explore racism in the less explored context of public health authorities. Several factors may explain this research gap, for example policies and practices distributing resources that do not prioritize health equity in public health services or research [30]. The lack of studies on racism in PHA can also be interpreted as epistemic injustice or violence [31]. Challenging this epistemic injustice in public health research must consider power relations and context-level factors in addressing health inequities [32–34]. We posit that PHA have the responsibility to acknowledge racism as a structural political determinant of health and a health hazard, and that racialization may occur even in their institution. They should monitor the effect of racism on community health. Furthermore, PHA should contribute to policymaking efforts in dismantling racism to ensure equitable health protection.

We mapped the scientific evidence on racism in public health authorities in a scoping review and interpreted it with a situational analysis (SA) [35]. We aimed to answer the following questions:

• How has racism or anti-racism in PHA been explored in the scientific literature?

• What are the experiences of racialized groups in PHA?

• In what kind of settings (e.g., interpersonal interactions, practices in public health services, discourses or policymaking) has racism been assessed in PHA?

• Did such assessments affect a) the PHAs' interaction with racialized groups; and b) health inequalities?

## Materials and methods

We conducted a scoping review, allowing for a broad but still comprehensive and systematic mapping of the evidence [36]. We aimed to include a diverse range of evidence, including implementation research on training programs as well as commentaries and opinion pieces as indicators of an ongoing discourse within the public health community. Additionally,

we aimed to assess how research on racism in PHA has been conducted. Thus, the approach of this scoping review was to assess the "nature and extent of research evidence" [36]. These aims are more compatible with a scoping review compared to a systematic review [37]. We followed the PRISMA-ScR checklist to ensure methodological and reporting quality [38] (see S1 Checklist) but did not publish a review protocol.

## Search strategy

We searched in the databases PubMed, Embase, PsycINFO and CINAHL. While we acknowledge that racialization shapes academic knowledge production and access to it, we excluded grey literature since this would have exceeded the scope of our review. We built our search strategy on three core elements: 1) racism as the experience and/or (in)action of interest, 2) PHA as the setting, and 3) the services provided by PHA or population health needs that often fall into the responsibility of PHA. For the racism element, we aimed to capture explicit forms and labels for racism, for example by including terms such as racialized harassment, microaggression or stereotypes, as well as intentions and actions to dismantle racism (e.g., anti-racism or racial equity) in the public health landscape. For the PHA setting, we used various synonyms from the literature that describe the institutions of different (country-)settings. In addition, we described the potential areas of action of the PHA building on the different roles and responsibilities illustrated in the background (e.g., health monitoring, exemplary infectious diseases such as HIV/AIDS or tuberculosis).

We developed the search strategy with the support of an experienced librarian. The search was conducted in June 2022, with an update on 4th October 2024 (see S1 Text for the full electronic search protocols). In addition to the databases, we conducted backward citation checks for the literature we included.

## Eligibility criteria and selection

We applied two major inclusion criteria: First, the publication must explicitly explore the setting of PHA. This includes any level from city to national or international and pertains the institution of PHA as a whole or their representatives. Second, the material needs to assess or discuss racism or anti-racism in actions, practices, guidelines, discourses, or the absence thereof, between any stakeholders involved. In other words, we included structural as well as inter-/intrapersonal forms of (anti-)racism. For example, this can encompass racism in interactions with service users or institutional guidelines or funding practices that racially discriminate population groups. We also included editorials, commentaries, viewpoints and comparable outlets since these give valuable insights in ongoing scientific discourses.

Accordingly, we excluded analyses of racism in healthcare facilities or racialized health inequalities without any explicit explanation of the role of PHA. The languages we were limited to in our research team are English, German, French, and Spanish. We did not apply temporal or regional exclusion criteria.

We conducted a pilot screening with a random sample of 5% of the total records to increase inter-rater reliability between the reviewers (LW, SS, TA). The team then discussed inconsistencies and adapted the eligibility criteria together. The reviewers then screened titles and abstracts. For the full texts, we again ran a pilot screening with 5% of the records identified in the title-abstract screening stage and then continued with the assessment. Together with the research team (YN, FOB), we resolved disagreements on the inclusion of potentially relevant papers.

## Data charting

The data charting comprised generic study-based information as well as data on 1) the PHA and their roles, 2) the racialized/anti-racist discourses and practices studied in the material, and 3) elements and actants for the situational analysis (S1 Table). These include SA specific aspects of analysis, for example individual, collective and implicated silenced human and non-human elements and actants (see next paragraph for further elaboration on SA analysis). We tested the

data charting with a random sample of 5 records (LW, SS, TA, FOB) to calibrate before use. Four reviewers then conducted the extraction independently and compared the data afterwards.

## Analysis

The analysis followed a two-step approach. First, we conducted a narrative synthesis guided by the data charting table. We provide descriptive statistics for the main characteristics of the material. For the second more analytical step, we used SA [35].

SA is an interpretative method for qualitative data, aiming to enrich constructivist grounded theory. SA centers on the broader situation as the primary unit of analysis which allows to capture the full complexity and messiness of social realities. By incorporating both human and non-human elements in its analysis (e.g., discourses) SA provides a more holistic view of the situation. SA employs different types of maps to analyze data to identify relevant human and non-human elements – often referred to as actants – in the social reality, to illustrate collective actors and their relationships and to represent different positions taken (or not taken) on issues within the social reality. SA is therefore particularly valuable to identify "sites of silences", actors and their relations as well as discourses. By emphasizing the inclusion and representation of visible and less visible actors, it allows to capture complex power structures. Thereby, SA is considered useful in addressing epistemic injustices, complementing the narrative approach in this scoping review.

Based on this approach, we created an ordered map which is typically used to identify the human and non-human elements relevant to the issue of inquiry, and what other elements may matter or affect the status quo. We used the classifying system as suggested by Clarke et al. [35] (see data charting form in S1 Table). While creating and analyzing the situational map, relationships and interconnections materialized that were highly relevant in interpreting the scientific discourse on racism in PHA and had implications for practice. Since we did not collect primary data in qualitative interviews, we did not memo these steps but did keep the different versions of the situational maps which can be accessed on request.

## Positionality statement

Our research team reflects a diversity of social positions, migration histories, and relationships to racism and marginalization. Our research focus and interpretive lens have been shaped by both lived experiences with racialization and exclusion within institutions and the commitment to reflexivity and allyship. Using SA has facilitated the recognition of the privileges associated with whiteness and positional power in public health structures. As part of SA, we engaged in ongoing reflection throughout the research process to examine how our positionalities influenced the framing, analysis, and interpretation of our work.

## Results

Of the 2704 records identified through our search strategy (after removing duplicates), we assessed 271 records for eligibility in the full text screening. We included 55 publications in our review after full-text screening and backward citation checks (Table 1). Details of the selection process are shown in Fig 1. The publications mainly comprised research articles (n = 33) and commentary pieces (n = 19). The research articles applied mostly qualitative methods [39–47], conducted reviews [48–58], document analyses [59,60] or were written in essay style [61–69]. Only two articles used quantitative data [70,71]. The publications were published between 1985 and 2024, more than half of them after 2021. Most originated in and/or explored the context of the US (n = 42).

### Racism in public health authorities

The health aspects assessed varied slightly by the group of PHA, but mostly pertained to infectious diseases such as sexually transmitted diseases, most prominently syphilis [50,53,65,66,69,72–75,77–80], HIV/AIDS

**Table 1. Characteristics of included studies, grouped by type of public health service involvements.**

| First author & year | Type of study | Study design | Region | Health topic | Assessment of (anti-)racism | Definition of racism | Explicit/ implicit | Participation of racialized groups | Perspective |
|---|---|---|---|---|---|---|---|---|---|
| **Abuse & violation of rights in scientific experiments (n = 20)** | | | | | | | | | |
| **Amster 2022 [72]** | Commentary | – | US | Syphilis | The Tuskegee Syphilis Study exploited African American men without consent, withholding treatment to observe disease progression, highlighting medical exploitation and racial marginalization. | – | Explicit | – | Retrospective |
| **Asabor 2021 [73]** | Viewpoint | – | US | Tuberculosis, syphilis | Medical mistreatment of Black individuals in the US public health services TB screening policies contribute to marginalization and exclusion of BIPOC and immigrant populations | – | Explicit | – | Retrospective & topical |
| **Czech 2023 [57]** | Research article | Review | Germany | Murder, involuntary sterilization | Public health authorities in Nazi Germany facilitated racial persecution by collecting hereditary data, enforcing eugenic policies, and supporting the exclusion of minorities. Their actions, including health surveillance and education campaigns, reinforced the systemic marginalization and persecution of Jews and other targeted groups. | "An important aspect of this past for contemporary medicine is that, despite the understanding of race as a social construct rather than an immutable biological fact, medical racism still exists.634 Medical racism is expressed not only in individuals' open bigotry and cruelty against racial and ethnic minorities, but also in subtle microaggressions and stereotypes held by people who would not consider themselves to be racist. Furthermore, systemic racism remains embedded in medical institutions and policies" | Explicit | – | Retrospective |
| **Dong 2022 [44]** | Research article | Qualitative | US | COVID-19 | Historical medical abuse is a major reference point for Black communities and affects COVID-19 vaccination uptake | No, but emphasis on structural elements | Explicit | Semi-structured interviews with Black Americans and stakeholders representing Black communities | Retrospective & topical |

*(Continued)*

| First author & year | Type of study | Study design | Region | Health topic | Assessment of (anti-)racism | Definition of racism | Explicit/ implicit | Participation of racialized groups | Perspective |
|---|---|---|---|---|---|---|---|---|---|
| **Feagin 2014 [50]** | Research article | Review | US | Syphilis | Medical mistreatment, ethical miscon- duct and eugenic experiments | Based on systemic racism theory " They argued that "racism" involves "predication of deci- sions and policies on considerations of race for the purpose of subordinating a racial group." While recognizing individual racism, they accented institutional (what we term systemic) racism that is "less overt" and "less identifiable in terms of specific individuals committing the acts. But it is no less destructive of human life."" | Explicit | – | Retro- spective & topical |
| **Lemelle 2003 [46]** | Research article | Qualitative | US | HIV/ AIDS | Medical mistreatment, ethical misconduct and eugenic experi- ments, PHA as part of a system criminalizing HIV | – | Implicit | – | Retro- spective & topical |
| **Miller 2007 [74]** | Book review | – | US | Syphilis | Medical mistreatment, ethical miscon- duct and eugenic experiments | – | Explicit | – | Retrospec- tive |
| **Nguyen 2021 [41]** | Research article | Qualitative | US | Contra- ception | Medical mistreatment, ethical miscon- duct and eugenic experiments | – | Explicit | Partnered with a community serving agency, stakeholders were involved in interviews | Retro- spective & topical |
| **Nuriddin 2020 [75]** | Commen- tary | – | US | Syphilis, repro- ductive health | Based on historical examples of racial injustices in medicine and the health care system in the US | – | Explicit | – | Retrospec- tive |
| **Patterson 2009 [65]** | Research article | Essay | US | Syphilis, infec- tious diseases | Medical mistreatment, ethical miscon- duct and eugenic experiments | No, but emphasis on structural elements | Explicit | – | Retrospec- tive |
| **Prather 2018 [53]** | Research article | Review | US | Syphilis, repro- ductive health | Medical mistreatment, ethical miscon- duct and eugenic experiments | – | Explicit | – | Retro- spective & topical |
| **Quimby 1993 [76]** | Commen- tary | – | US | HIV/ AIDS | Systemic exclu- sion from research protocols | – | Implicit | – | Retro- spective & topical |

*(Continued)*

**Table 1.** (Continued)

| First author & year | Type of study | Study design | Region | Health topic | Assessment of (anti-)racism | Definition of racism | Explicit/ implicit | Participation of racialized groups | Perspective |
|---|---|---|---|---|---|---|---|---|---|
| **Randall 1996 [66]** | Research article | Essay | US | Syphilis | Medical mistreatment, ethical misconduct and eugenic experiments | – | Explicit | – | Retrospective |
| **Reverby 2011 [77]** | Perspective | – | US | Syphilis | Medical mistreatment of Black individuals in the US public health services | – | Explicit | – | Retrospective |
| **Reverby 2016 [78]** | Editorial | – | US, Guatemala | Syphilis, STD | Medical mistreatment of Black individuals in the US and Guatemalan public health services Call for "reconciliation, [...] compensation, and restorative justice" | – | Explicit | – | Retrospective & topical |
| **Reverby 2020 [79]** | Position paper | – | US, Guatemala | Syphilis, STD | Medical mistreatment of Black individuals in the US and Guatemalan public health services Call for reparations and compensation | – | Explicit | – | Retrospective & topical |
| **Reverby 2022 [80]** | Commentary | – | US | Syphilis | (Historical) medical mistreatment of Black individuals in the US public health services. PHA acquired funding from the New York's Milbank Memorial Fund to be able to provide "burial insurance" to keep the men in the study. | – | Explicit | – | Retrospective |
| **Riley 2022 [60]** | Research article | Policy analysis | US | Reproductive health | The lack of funding for abortion-related research and gaps in surveillance data in the U.S. reinforce White supremacy and hinder equitable access to abortion, while public health systems could play a key role in improving access. Historically, reproductive health fields have contributed to state-sanctioned eugenics and forced sterilizations, particularly targeting Indigenous women. | "Structural racism refers to the "state-sanctioned and/or extralegal production and exploitation of group-differentiated vulnerability to premature death"4(p28) that works through "mutually reinforcing inequitable systems."5(p1454) Structural racism is sustained through White supremacy, which is the system of conditions and ideologies that underscore the hegemony of whiteness and White political, social, cultural, and economic power." | Explicit | – | Topical |

*(Continued)*

**Table 1.** (Continued)

| First author & year | Type of study | Study design | Region | Health topic | Assessment of (anti-)racism | Definition of racism | Explicit/implicit | Participation of racialized groups | Perspective |
|---|---|---|---|---|---|---|---|---|---|
| **Stern 2005** [68] | Research article | Essay | US | Reproductive health (sterilizations) | Among other public institutions, the Department of Health, Education and Welfare (HEW) backed about 100 000 sterilizations annually, with mostly African American women being involuntarily sterilized | – | Implicit | – | Retrospective |
| **Tobin 2022** [69] | Research article | Essay | US, Guatemala | Syphilis, gonorrhea | Medical mistreatment of Black individuals in the US and Guatemalan public health services | "For years, researchers have treated race as an innate genetic attribute, whereas the perspective of race as a social construct is now widely embraced. The term "structural racism" is used to convey that racism has a systemic basis, embedded in social policy and norms and not simply private prejudices of individuals." | Explicit | – | Retrospective |
| **Constructing groups based on race & ethnicity - Othering practices (n = 8)** | | | | | | | | | |
| **Galis-hoff 1985** [61] | Research article | Essay | US | Mortality (infectious diseases) | Racist and eugenic discourses against Black communities, upheld and disseminated by public health officials | – | Explicit | – | Retrospective |
| **Gee 2011** [51] | Research article | Review | US | Infectious diseases | By constructing immigrants as diseased Others, PHA reinforced racial stereotypes which was also used as a reference and justification for restricting immigration and white-only citizenship | "Structural racism is defined as the macrolevel systems, social forces, institutions, ideologies, and processes that interact with one another to generate and reinforce inequities among racial and ethnic groups (Powell 2008)." | Explicit | – | Retrospective & topical |
| **Gelpí-Acosta 2022** [81] | Commentary | – | US | HIV, drug abuse | The lack of disaggregated health surveillance data within racial communities hinders targeted prevention efforts, particularly for subgroups made structurally vulnerable like Puerto Ricans facing higher risks for HCV and HIV. | – | Implicit | – | Topical |

*(Continued)*

| First author & year | Type of study | Study design | Region | Health topic | Assessment of (anti-)racism | Definition of racism | Explicit/ implicit | Participation of racialized groups | Perspective |
|---|---|---|---|---|---|---|---|---|---|
| **McMillen 2021 [82]** | Editorial | – | Canada | Tuber-culosis | Racialized framing of tuberculosis infection in Indigenous people | Racial susceptibility theory & virgin soil theory | Implicit | – | Retrospec-tive |
| **Moseby 2013 [40]** | Disserta-tion | Qualitative | US | HIV/ AIDS | Inclusion & exclusion of black Americans in HIV/AIDS surveillance and prevention inter-ventions by the CDC | – | Explicit & implicit | Interviews with Black Ameri-cans from CDC and community organizations | Retro-spective & topical |
| **Proctor 2011 [42]** | Research article | Qualitative | Nether-lands | HIV/ AIDS | Constructing ethnic minorities as the "high-risk sexual other" | No, but definitions of race, ethnicity and immigration status have been discussed (socially constructed concepts vs. biological conceptualization) | Explicit & implicit | – | Topical |
| **Roberts 2009 [67]** | Book | Essay | US | Health dispar-ities, tubercu-losis | Constructing African American as diseased Others, thus justify-ing segregation and isolation policies in particular for tubercu-losis management in the 19/20th century | No, but race is described "less in terms of supposed truths of biological dif-ference (including skin color) and more in terms of the multiple ways in which differ-ence and inequality may be articulated, mobilized, and experi-enced within dynamic political and economic systems." | Explicit | – | Retrospec-tive |
| **Oyola-Santiago 2024 [83]** | Commen-tary | – | US | Drug abuse, over-doses | A Latinx community-driven campaign (Narcanazo) advo-cated successfully for disaggregated data and analysis in public health agencies of the Latinx concept in drug abuse and overdose surveillance. | – | Explicit | The campaign was devel-oped and implemented by community members, (also reflected in authorship of the study) | Topical |
| **Guidelines, recommendations & policy making (n = 11)** | | | | | | | | | |
| **Abelson 2024 [84]** | Commen-tary | – | US | Covid-19 health inequali-ties | The Washington State Department of Health developed a funding methodology prioritizing equity by integrating community voices and quantita-tive criteria to allocate COVID-19 resources. | – | Implicit (they used a racial equity lens and prioritized intersec-tionalities in commu-nities) | Engaged com-munity orga-nizations at a grassroots level and established sustainable partnerships to design and implement the methodology | Topical |

*(Continued)*

| First author & year | Type of study | Study design | Region | Health topic | Assessment of (anti-)racism | Definition of racism | Explicit/ implicit | Participation of racialized groups | Perspective |
|---|---|---|---|---|---|---|---|---|---|
| **Bonacci 2021 [48]** | Research article | Review | US | HIV/ AIDS | Racial/ethnic disparities in PrEP care and associated strategies to address these inequities | No, but emphasis on structural elements | Implicit | – | Topical |
| **Bungay 2023 [85]** | Research article | Document review | Canada | Sex work-related health | Racism is neglected in sex work related health research funded by the federal funding agency in Canada | No, but emphasized the systemic inequities | Explicit | – | Topical |
| **Came 2014 [43]** | Research article | Qualitative | New Zealand | Public health policy making | Suppressing Maori voices and knowledge in policy making processes | Focus on institutional racism against Maori: "the outcomes of mono-cultural institutions which simply ignore and freeze out the cultures of those who do not belong to the majority. National structures are evolved which are rooted in the values, systems and viewpoints of one culture only. Participation by minorities is conditional on their subjugating their own values and systems to those of "the system" of the power culture." (Ministerial Advisory Committee on a Maori Perspective on Social Welfare, 1988, p. 19) | Explicit | Storytelling and participatory analysis, epistemological frameworks | Topical |
| **Lowe 2022 [63]** | Research article | Essay | UK, US | Contraception | Disproportionate recommendations for and use of long-acting reversible contraception (LARC) among Black people compared to white people | No, but emphasis on structural elements | Explicit | – | Retrospective & topical |
| **Manca 2021 [39]** | Research article | Qualitative | US | COVID-19 | Framing of public health recommendations and differing impact on social positions (e.g., BIPOC) | – | Explicit | – | Topical |
| **McRoy 2007 [64]** | Research article | Essay | US | Policy making | Delay or denial of foster care or adoptive placement among ethnic minority children | No, but emphasis on structural elements | Implicit | – | Retrospective & topical |

*(Continued)*

| First author & year | Type of study | Study design | Region | Health topic | Assessment of (anti-)racism | Definition of racism | Explicit/ implicit | Participation of racialized groups | Perspective |
|---|---|---|---|---|---|---|---|---|---|
| **Mota 2024 [59]** | Research article | Historical analysis | Brazil | Sickle cell disease | Historical neglect of sickle cell disease and the associated racial inequalities reflect institutional racism | – | Explicit | – | Retro-spective & topical |
| **Parmet 2007 [86]** | Commen-tary | – | US | Tuber-culosis, plague | Disproportionate use of isolation against marginalized, Black & PoC people | – | Implicit | – | Retro-spective & topical |
| **Yates 2023 [54]** | Research article | Imple-mentation research | US | mater-nal and child health | Incorporating Repro-ductive Justice into state-funded programs of long-acting revers-ible contraception and reproductive-life planning. | No, but racism is conceptualized as one axis of inequality in reproductive justice | Explicit | SisterSong Women of Color Repro-ductive Justice Collective and National Women's Health Network adjusted the evidence based strategy | Topical |
| **Legal obligation to tackle health inequities (n=4)** | | | | | | | | | |
| **Chin 2018 [49]** | Research article | Review | New Zea-land, US | Health dispari-ties | Emphasizing the need for equity-driving pol-icy making, including the acknowledgement of systemic racism, to effectively reduce health disparities | – | Explicit | – | Retro-spective & topical |
| **Kereama-Royal 2019 [87]** | Commen-tary | – | New Zealand | Breast cancer | Lack of Maori-specific support and guidelines for cancer treatment | "A pattern of behav-ior that benefits one ethnic group and dis-advantages another. Institutional racism can manifest through policy, investment decisions, mono-cultural structures and inaction. Critically, racism does not need to be intentional." | Explicit | Written by Maori & non-Maori researchers | Topical |
| **McSpedon 2022 [88]** | Commen-tary | – | US | Health dispari-ties | Racism manifested in health inequalities | – | Explicit | – | Topical |
| **Pillaye 2020 [89]** | Corre-spon-dence | – | UK | COVID-19 | Discriminatory actions and historically racialized inequalities (working conditions & health outcomes) | – | Implicit | – | Topical |

*(Continued)*

| First author & year | Type of study | Study design | Region | Health topic | Assessment of (anti-)racism | Definition of racism | Explicit/ implicit | Participation of racialized groups | Perspective |
|---|---|---|---|---|---|---|---|---|---|
| **Surveillance and reporting (n = 14)** | | | | | | | | | |
| **Amani 2023 [55]** | Research article | Review & qualitative interviews | US | Health equity, COVID-19 | An equity-based scoring system was developed to evaluate and mitigate the harms of public health surveillance on communities of color, focusing on risks from policing and data misuse. | – | Explicit | – | Topical |
| **Bertolli 2007 [70]** | Research article | Quantitative | US | HIV/ AIDS | Racial misidentification in HIV/AIDS reporting systems leading to undercounting | – | Implicit | – | Topical |
| **Csete 2023 [56]** | Research article | Review | Canada, US | HIV | Molecular HIV surveillance systems, while mandated for federal funding, can perpetuate discrimination and privacy risks, disproportionately harming marginalized groups. | – | Explicit | – | Retrospective & topical |
| **Daroya 2023 [90]** | Research article | Mixed-methods | Canada | COVID-19 | COVID-19 restrictions, disseminated by Public Health Officers, are criticised to fail to account for heterogeneity of queer people's experiences of homelessness and structural racism. | No, but explained that they use the term "racialized people" to emphasize that race is a social construct | Explicit | As interviewees | Topical |
| **Decoteau 2022 [58]** | Research article | Review & qualitative interviews | US | COVID-19 | Chicago's focus on disease surveillance over social safety nets hindered efforts to achieve racial equity during COVID-19, disproportionately affecting Black and Latinx communities. The Racial Equity Rapid Response Team (RERRT) collaborated with officials and community groups to address these disparities through targeted testing, contact tracing, and community support initiatives. | – | Explicit | – | Topical |
| **Garcia 2022 [62]** | Research article | Essay | US | COVID-19 | Different effects and reactions to surveillance measures during the COVID-19 pandemic in Black communities compared to white communities | No, but emphasis on structural elements | Implicit | – | Topical |

*(Continued)*

**Table 1.** (Continued)

| First author & year | Type of study | Study design | Region | Health topic | Assessment of (anti-)racism | Definition of racism | Explicit/ implicit | Participation of racialized groups | Perspective |
|---|---|---|---|---|---|---|---|---|---|
| **Grineski 2006 [45]** | Research article | Qualitative | US | Tuber-culosis | Dehumanizing and exclusionary discourses in public health reporting and recommendations to contain the spread of TB | – | Explicit | – | Retrospec-tive |
| **Harp 2020 [52]** | Research article | Review | US | Drug abuse reporting | Racial disparities in surveillance and reporting of drug use in Black pregnant women | – | Implicit | – | Topical |
| **Hastings 2021 [91]** | Commen-tary | – | US, Canada | COVID-19, HIV/ AIDS | Disproportionate surveillance and law enforcement of Black people compared to white people | No, but emphasis on structural elements | Explicit | Indirectly by referencing Indigenous voices | Topical |
| **Marteau 2005 [92]** | Editorial | – | UK | Sickle cell screen-ing | The failing of public (health) institutions to respond to ethnic minority populations' needs | "Failings of public institutions to respond to the needs of ethnic minority populations […] Institutional racism is in effect the uncritical application of policies and proce-dures that ignore the needs of an ethnically diverse society. Such practices, by default, favor the white population." | Explicit | – | Topical |
| **Mody 2021 [71]** | Research article | Quantitative | US | COVID-19 | Racial disparities in COVID-19 testing per case as a measure for health inequity | No, but emphasis on structural elements | Explicit | – | Topical |
| **Molldrem 2023 [93]** | Commen-tary | – | US | HIV | Predictive analytics in HIV surveillance systems require new approaches to data ethics, rights, and reg-ulation in public health to avoid the risk of mis-classifying people and further criminal-ization of HIV. | – | Yes | – | Topical |
| **White 2023 [47]** | Research article | Qualitative | US | Racism as a social determi-nant of health | There is a paucity of Public Health Surveillance Sys-tems that measure individual-level rac-ism, and few systems are linked to structural racism measures. | – | Explicit | – | Topical |

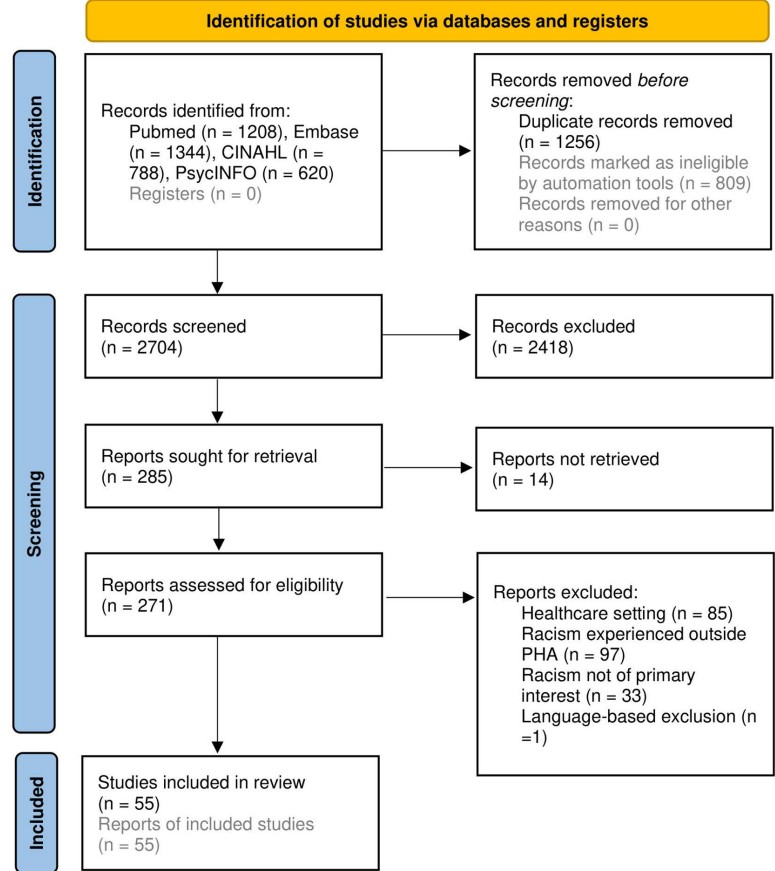

**Fig 1. Flowchart on the selection of sources of evidence.**

[40,42,46,48,56,70,76,81,93], Covid-19 [39,44,55,58,62,71,76,84,90,91], or tuberculosis [53,67,73,75]. Reproductive health including contraception [41,53,54,60,63,68,75,85] was a recurring theme, as were more generic discussions of health inequalities [47,49,88].

Only eight publications explicitly defined the term racism in their work [43,50,51,57,60,69,87,92]. These definitions all emphasized the institutional and structural characteristics of racism, relating the processes of categorizing and devaluing groups, manifestations in societal structures ultimately resulting in power hierarchies putting racialized groups at a disadvantage. In addition to these characteristics, Came [43] specifically defined institutional racism against Māori while Feagin and Bennefield [50] built on the systemic racism theory. While most of the publications lacked an explicit definition, some provided definitions for concepts like race, ethnicity and (im)migration history [42,67,90], different ideologies of racial thinking (i.e., racial susceptibility and virgin soil theory) of the 20th century in the US [82] or emphasized the structural character of racism in their work [44,48,54,62–65,85,91]. Thirty-three publications did not give definitions of any of the concepts.

The racialized discourses, practices or interactions in PHA can be broadly categorized in terms of five different roles according to the public health services involved: 1) conducting scientific experiments (n = 20 [41,44,46,50,53,57,60,65,66,68,69,72–80]); 2) surveillance and reporting (n = 14 [40,45,47,52,55,56,58,62,70,71,90–93]); 3) designing and implementing guidelines, recommendations and policymaking (n = 11 [39,43,48,54,59,63,64,84–86,89]);

4) practices of constructing groups based on racialization and Othering (n = 8 [40,42,51,61,67,81–83]); and 5) the legal obligation to tackle health inequities (n = 4 [49,87–89]).

In the first category, most studies in this realm elaborate on and criticize medical abuse of African American men in the infamous Tuskegee Syphilis experiment conducted by the US Public Health Services between 1932–1972 [41,44,46,50,53,66,69,72–74,77–80]. In a nutshell, PHA studied African American men to observe the course of disease without informed consent. Treatment that became available later was knowingly withheld and actively suppressed. Such violations are not only specific to the US context but have also been committed in Guatemala by the US Public Health Services, though not over such a long period of time [69,78–80]. Only one study reviewed the role of PHA in Nazi Germany, facilitating racial persecution by collecting hereditary data and enforcing eugenic policies [57]. In addition to these atrocities, the publications also illustrate PHA as being complicit in forced sterilizations mostly targeting economically disadvantaged African American women (e.g., through providing funds) [68] and how abortion criminalization and lack of reporting thereof can be interpreted as a continuation of controlling reproductive health [60]. One study elaborates on how PHA were responsible for designing research protocols, monitoring practices and laws that systematically exclude Black people and risk to legitimize the exploitation of Black people's bodies for medical purposes [76]. Given the thematic focus on past events, all publications applied a retrospective lens in their analyses with few also exploring their relevance and impact on today's experiences in racialized groups. The PHA acted both on a regional and national level and pertained to racialized groups in the US, Guatemala and Germany, mostly referred to as African American or Black (and POC) communities. Only two studies included voices of racialized groups through semi-structured interviews [44] and partnering with a community serving agency [41].

The surveillance and reporting category displays multiple sites of racism and racialized practices. These pertain to racial disparities in accessing screening services for COVID-19 or sickle cell screening [71,92], and racial misidentification in HIV/AIDS reporting systems leading to undercounting [70]. The authors found several containment measures that they identified as dehumanizing and exclusionary discourses in public health reporting [45]. In addition, multiple publications elaborated on the racialized impacts of surveillance measures in Black communities during the COVID-19 pandemic [58,62,90] as well as disproportionate screening for drug use in Black pregnant women [52]. Three studies criticized the risks of using public health surveillance data in law enforcement and policing, e.g., for HIV-criminalization [55,56,93]. One study reviewed (the lack of) measures of racism as a social determinant of health in public health surveillance systems [47]. With regards to the perspective, all studies but one [45] explored recent issues at the national and regional level of the respective country. Participation of racialized groups was not realized, only two referenced Indigenous voices explicitly [91] or involved racialized groups as interviewees [90].

The third category depicts how racialized groups can be disproportionately targeted [63,86], put at a disadvantage by public health policies and guidelines [39]; or not being heard or included in the development of such [43]. In addition, PHA can advocate and influence policies – most effectively with community-based organizations - to increase equity and reproductive justice aspects in funding practices [54,84,85]. This also pertains to related fields, i.e., child welfare services, to ensure racial equity in foster care and adoptive placement in the US nowadays [64]. A common thread among these publications is the demand to take into account the needs of racialized groups and to prevent disparate impact in the first place. Policies on COVID-19 and HIV/AIDS PrEP care naturally evolve topical issues, while policies on contraception [63], tuberculosis [86], sickle cell disease [59] and child welfare services [64] also draw on historical events to help explain today's patterns. The policy-making processes encompass regional, state and national level PHA. Came used storytelling and participatory analysis as well as Indigenous epistemological frameworks in their study, displaying the highest level of participation of racialized groups in this review (and the only participatory approach for this category) [43].

PHA constructed high risk groups, especially for infectious diseases, based on race or ethnicity to justify spatial segregation and isolation policies to contain the spread in Indigenous, African American and ethnic minorities [61,67,82] or to support restricting the immigration of these groups [51]. While these studies reflected on historical practices, Proctor et al.

show how ethnic minorities are still constructed more recently as "high-risk sexual others" in HIV/AIDS reporting in the Netherlands [42]. They also illustrate that the discourses upheld and constructed by PHA translate into media and politics, perpetuating racialized stereotypes and Othering even more. Also, these framings can affect whether racialized groups have access to services [40]. On the other hand, public health data are blind to subgroups made structurally vulnerable with regard to drug abuse due to a lack of disaggregated data falsely homogenizing, e.g., the Latinx community [81,83]. In terms of representation, one study involved voices of Black Americans to explore the inclusion and exclusion in HIV/AIDS prevention services provided by the CDC [40], while community-organizations were driving forces in designing and implementing a campaign for disaggregated data in public health agencies [83].

In the fifth and last category, the publications explored and discussed how well PHA fulfil their legal obligation to dismantle health inequities. These range from acknowledging that racism manifests in health inequities [88], the lack of Māori-specific [87] or even discriminatory policies in New Zealand/Aotearoa [89] to the impetus to finally build on equity-driven principles in policy-making [49]. The commentary on Māori-specific needs in breast cancer care is written by Māori and non-Māori researchers and policymakers, representing again the only study with participatory elements. All publications address the current deficits in policies to address health inequities, and only one links these with past events [49]. PHA operate on the national level in this category, while Kereama-Royal also takes regional policies into account [87].

## Situational analysis of discourses

The ordered map (Table 2) provides a structured overview of discourses and discursive constructions surrounding racism and anti-racism in PHA. It identifies how systemic and institutional racism, alongside broader sociopolitical structures, shape the practices and policies of PHA. Discourses are framed around key human and non-human actants, with marginalized racialized groups and public health authorities taking center stage in debates over inequities, surveillance, and justice. SA also suggests racialized groups may be implicated or silenced groups in racism research surrounding PHA. Very few studies include positionality statements which makes it difficult to conclude whether racism research also constructs discourses with or about racialized groups. In only a few of the publications, racialized groups were also vocal in the knowledge production and design of practices in PHA.

Racialized communities, including Black, Indigenous, Latinx, and Asian populations, tend to be constructed as collectives, "high-risk others" (e.g., [42]) within public health discourses, perpetuating exclusionary narratives and structural inequities. Historical abuses like forced sterilizations and unethical research practices (e.g., Tuskegee Syphilis Study) further entrench mistrust and highlight the complicity of PHA in sustaining racial hierarchies.

PHA and their policies often prioritize surveillance and risk management over structural interventions, reflecting broader colonial, racial, and economic power imbalances. Reporting systems and guidelines frequently neglect the needs of racialized populations, while practices such as targeted testing and policing exacerbate health inequities and perpetuate stereotypes. These discursive constructions, rooted in White supremacy, ignore the agency of marginalized communities and fail to engage them in meaningful ways, further deepening disparities.

Counter-discourses that advocate for transformative approaches are constructed by collective research actors explicitly informed by Reproductive Justice, critical race theory, decolonization and anti-racism frameworks. These perspectives emphasize the importance of centering marginalized voices, engaging community leaders, and co-creating policies to address systemic violence and promote equity. PHA are called to confront structural racism explicitly and implement equitable, community-driven interventions.

Central themes in discourses include systemic and institutional racism, the perpetuation of inequities through white racial framing and colonial legacies, and the role of public health in reinforcing or challenging these structures. Related discourses draw from critical race theory, anti-racism scholarship, and Reproductive Justice frameworks, emphasizing the intersectionality of race, gender, class, and colonial histories. Counter-discourses also advocate for equity-centered approaches that address structural violence and prioritize inclusive, community-led strategies to dismantle systemic

**Global Public Health**

**Table 2. Ordered map on racism and anti-racism in public health authorities – an overview of elements, actants, discourses and discursive constructions.**

| Individual human elements/actants | Nonhuman elements/actants |
|---|---|
| <u>Researchers with a background/affiliation to the following disciplines</u><br>• Within public health: medicine, epidemiology, population health, public health, health ethics, nursing, healthcare law,<br>• Psychology, historical sciences, social sciences, social work, sociology, anthropology, law, liberal studies, women's and gender studies, feminist studies<br><u>Few researchers revealed their ethnicity:</u> 7th generation Pakeha (settler) New Zealander with a background in public health and anti-racism activism [43], Researchers are Māori and non-Māori breast cancer survivors (and allies) [87], Asian American roots (at least primary investigator) [41] | <u>Reporting systems</u><br>• Cancer reporting system, Covid-19 control measures, Tuberculosis/ HIV/ AIDS, Sickle cell screening programs<br><u>Guidelines & policies</u><br>• *Public health specific:* Drug abuse, long-acting reversible contraception, equitable PrEP care, health equity, national reporting systems, sterilization laws<br>• *Collaboration with other social services:* Policing of HIV/AIDS status, adoption and foster care, Interethnic Placement Act (IEP), Immigration policies and citizenship requirements |
| **Collective human elements/actants** | **Implicated silent elements/actants** |
| Public health authorities (including agencies on the international level - WHO, PHAC; national level – CDC, NHS, National Institute for Public Health and Environment; and regional and city level offices) as well as their representatives/officials<br>Community-rooted organizations and public health experts focused on addressing inequities.<br>• Reproductive Justice and HIV-focused organizations<br>• Black feminists and reproductive justice scholars | <u>Racialized groups:</u><br>• Indigenous Communities (American Indians/Alaska Natives, Indigenous people in Canada, Maori health leaders and women)<br>• Black and African American Communities:<br>• Hispanic/Latinx Communities<br>• Asian and Middle Eastern Communities:<br>• Specific Marginalized Groups: LGBTQ community, Sex workers, People with disabilities or psychiatric illnesses, Political prisoners, prisoners of war, and others historically marginalized.<br>Researcher positionality: Few researchers reveal racialization/ ethnic background |
| **Discursive constructions of individual and/or collective actants** | **Discursive constructions of nonhuman actants** |
| <u>Barriers and Systemic Inequities</u><br>• Black and Latinx communities face systemic barriers in accessing healthcare, including PrEP and HIV care, compounded by institutional neglect.<br>• Racialized groups, including Indigenous, Black, and ethnic minorities, are framed as "high-risk others" and face systemic health inequities rooted in systemic racism and White supremacy within public health institutions.<br>• Immigrants and ethnic minorities are often portrayed as the "diseased Other," reinforcing exclusionary narratives.<br>• Persistent neglect of diseases prevalent among marginalized groups (e.g., sickle cell disease in Brazil) reflects structural racism in public health systems.<br><u>Community Leadership and Advocacy</u><br>• Community-driven initiatives (e.g., Narcanazo in Puerto Rican communities) highlight the importance of culturally specific interventions to address health crises like overdoses.<br>• Engaging marginalized communities and their leaders in public health policy-making and funding allocation ensures culturally relevant and ethical practices.<br><u>Historical and Ongoing Injustices</u><br>• Institutional racism underpins historical abuses, such as forced sterilizations of African American, Puerto Rican, and Native American women, and continues to shape modern public health practices.<br>• Black pregnant women are disproportionately subjected to drug screenings and surveillance, reinforcing stereotypes of unfit parenting.<br>• African Americans living with HIV face criminalization and discriminatory practices in public health and law enforcement. | <u>Critiques of Public Health Authorities and Surveillance Systems</u><br>• Their narratives are interpreted and framed as a continuation of colonial, racial and economic power imbalances and/or the manifestation of institutional racism<br>• Focus on risk management and individualization of structural inequalities (e.g., in exposure to precarious living conditions and access to healthcare and insurance)<br>• Supported, funded and designed racist and eugenic interventions in the name of public health (Tuskegee, Guatemala, Nazi-Regime)<br>• Neglect, ignore or do not assess the needs of racialized populations groups sufficiently<br>• Perpetuate racialized stereotypes and Othering in disproportionate surveillance, regulations or interventions<br><u>Policing and Surveillance</u><br>• Policing within public health increases violence, mental distress, incarceration, and denial of services for Black, Indigenous, and People of Color (BIPOC).<br><u>Recommendations for Equity</u><br>• Public health officials must acknowledge systemic racism as a determinant of health inequities and prioritize equity-driven, community-led approaches.<br>• Ethical public health interventions require centering the voices of affected communities, particularly in policy design and implementation.<br>• Reproductive Justice frameworks and critical race praxis should guide public health efforts to address structural violence and health disparities. |
| **Political/economic elements** | **Sociocultural/symbolic elements** |
| • Healthcare system (prevention, curative and rehabilitative services), e.g., National Health Service (NHS), Medicaid<br>• Public health associations & organizations<br>• Social services including social work, child welfare services,<br>• Government and Policy-Making Bodies: Federal authorities, political parties, city governments (e.g., Chicago), and policymakers<br>• Law Enforcement and Criminal Justice Systems, incl. Immigration (control) systems<br>• Military forces<br>• Media | • Race & ethnicity (in almost every record, often a combination of both without explicit differentiation)<br>• Immigration history, nationality and citizenship, languages spoken, place of birth<br>• Sex/gender (without differentiation), LGBTQ+ identities, sexual orientation and/or attraction, motherhood/parenthood<br>• Socioeconomic resources, education<br>• Region of residence, time<br>• Age<br>• Health status (HIV/AIDS, STI infection, chronic diseases), (dis-)ability status |

*(Continued)*

**Table 2.** (Continued)

| Temporal elements | Spatial elements |
|---|---|
| Historical Periods - 19th century to today, including:<br>• Slavery, Black Codes, Jim Crow era, segregation, civil rights, and post-civil rights era.<br>• 1930s-1970s Tuskegee Syphilis experiment<br>• Holocaust period<br>Topical periods, with specific mention of:<br>• COVID-19 Pandemic<br>• Post-Roe era | • US<br>• UK<br>• New Zealand<br>• Canada<br>• Guatemala<br>• Netherlands<br>• Germany |
| **Major issues/debates (usually contested)** | **Related discourses (historical, narrative, and or visual)** |
| Structural and Systemic Racism:<br>• Systemic racism, institutional racism, structural racism.<br>• White racial framing, implicit bias, white saviorism, racial equity.<br>• Racism in healthcare, medical mistrust, and health inequities.<br>• Racism in public health policies, practices, and narratives (e.g., deprivation and inequality).<br>• Health Equity and Social Justice, Social Determinants of Health as counter-acting debates<br>Critical Race and Intersectionality:<br>• Critical race theory, anti-racism scholarship, and intersectionality.<br>• Knowledge production, patriarchy, heteronormativity, gender equality<br>Reproductive Justice and Gender:<br>• Reproductive rights, sexual autonomy, birth control, and abortion rights.<br>• Misogyny, "ideal" parenthood, intensive motherhood ideologies.<br>• Forced sterilization, contraception, and the criminalization of marginalized groups.<br>• Sex work-related stigma<br>• Stigma, xenophobia, and health inequities in HIV/AIDS care and prevention.<br>Criminalization and Surveillance:<br>• Racism in criminal justice: criminalization, mass incarceration, and HIV stigma.<br>• Surveillance and targeted testing perpetuating stereotypes and inequities.<br>• Policing and structural violence as systemic barriers. | Colonialism and Decolonization:<br>• Colonialism, decolonization, and the ongoing impacts of European colonization.<br>• Kaupapa Māori theory and Indigenous perspectives (e.g., New Zealand context).<br>• Othering and epistemic dominance in colonial and post-colonial settings.<br>Historical and Ethical abuse:<br>• Medical apartheid, medical experimentation, and unethical research.<br>• Anti-Semitism, Nazism, and the Holocaust.<br>• Eugenics, segregation, and ethical abuse in medicine and public health.<br>• Historical narratives: Jim Crow era, Reconstruction, and racial violence.<br>• Germ theory, sanitation, and racist ideologies in historical public health.<br>Economic and Political Factors:<br>• Neoliberalism, capitalism, and economization in public health.<br>• Structural violence embedded in political and economic systems.<br>• Racial inequities exacerbated by economic exploitation and marginalization. |

racism within PHA. These debates reflect a struggle between maintaining status quo power dynamics and advancing justice-driven public health interventions.

## Discussion

We identified a broad range of areas of public health services that are relevant to and connected with PHA. The different areas include the conduct of scientific experiments, different areas of surveillance, designing and implementing guidelines, recommendations, and policymaking, the construction of groups based on race, ethnicity or nationality in health reporting, as well as the fulfilment or neglect of the legal obligation to tackle health inequities. Health promotion was not explicitly addressed as an area of public health services. The included publications underline the structural and institutional character of racism. For example, public health data have been criticized to falsely homogenize and thus being blind to marginalized populations – such as the Latinx community, which leads to systemic disadvantages and discrimination in monitoring, reporting and funding practices of PHA. The fact that retrospective analyses dominate within the extracted publications leads to the argument that for understanding PHA and their practices, a historical perspective is indispensable. Moreover, it highlights how past events and experiences still have an impact on PHA practices and their interaction with racialized groups today. Even though we found a highly context-specific relationship between experiences and practices in relation to racism with regard to time and place, a global history of racism seems to be embedded overall.

Within the framework of the included studies, we did not find evidence of intrapersonal attitudes of people working in PHA or interpersonal interactions between PHA and racialized groups – contrary to research focusing on the health care setting where these perspectives are often core to the analysis [94–96]. In our review, the studies mostly take on a macro perspective that analyzes structural- and system-level practices or policies. Thus, tackling practices and policies at these levels is an important step to dismantle racism in public health institutions, even though most of them are described with a retrospective lens. Indeed, the relative lack of research into ongoing or very recent incidences of racism in PHA is noteworthy, pointing perhaps to both the "benefit of hindsight" but also the ethical challenges and power difficulties in "exposing" racism in ongoing policies and practice within PHA.

Based on our findings, direct interactions between PHA and racialized groups seem to be "a site of silence," that is, an unarticulated part of the situation/social reality according to SA terminology, in the literature on PHA. Even if some of the studies acknowledge that the identified barriers/challenges/inequalities are the direct consequence of structural racism they did not account for or directly assessed experiences of racism in their analyses, e.g., [97], supporting our identification of a site of silence.

Reproductive health, including abortion and contraception, as well as sexually transmitted diseases like HIV/AIDS and syphilis were prominent health outcomes in our findings. On the one hand, these areas are associated with high level of regulations and potentially criminalization – from a historical perspective until today [27,98]. On the other hand, these fields highlight the intersecting systems of oppression next to racism, that shape unique experiences in socially marginalized populations, such as women, LGBTQ+ [5,8]. Specifically, it is worth pointing out that racialization in a health care setting can sometimes go hand in hand with (hyper-) sexualization, often linked to 'exotic' cultural imagery emerging from historical slave trade, contemporary and sex trafficking [99], all of which warrants further dismantling in the context of anti-racist PHA reproductive health services in the future.

While we document historical and present facets of racism in PHA, our review also provides positive examples of PHA acknowledging and taking actions to dismantle racism in their practices and services. These highlight the importance of a) building the services in PHA on anti-racism and equity-driven principles [49,54,55] and b) integrating and amplifying the voices of racialized community [54,83,84]. Furthermore, these studies also take racialized positionality into account or were conducted by racialized researchers themselves. The examples illustrate how this enriches and improves the broad range of services, including resource allocation for COVID-19 response [84] and funding for reproductive health services [54], disaggregating data [83] and equity-based scoring systems for surveillance matters [55] as well as strategies to reduce racial inequities in prevention services [48]. While these can serve as valuable starting points, also in the sense of restorative justice [78], PHA need to be careful to not put the burden of un-doing racism on racialized groups alone. Additional approaches of PHA aiming to reduce racial health inequities as well as racist practices in surveillance systems, for example from Germany, can be based on diversity or culturally sensitive approaches [100–102].

## Strengths and limitations

To our knowledge, this review is the first to shed light on PHA as an underexplored area but yet highly relevant in terms of institutional power and its role in the protection of the population's health. We also believe that this review is the first of a kind in terms of employing SA as an analytical strategy. We systematically and comprehensively mapped the scope of research and scientific debate while also capturing how research has been conducted. Complementing the narrative review approach with situational analysis as a tool to highlight the relations and interconnectedness of human and non-human elements shaping racialized events and experiences in the setting of PHA is another strength of this review. Moreover, we contribute to restoring epistemic justice and highlight silenced groups without reinforcing by linking situational analysis with an analytical lens informed by decolonial scholarship.

We decided to only map the literature that captures explicit reference to racism or racial discrimination indexed in the most relevant scientific databases in public health research. This is why we cannot claim to give a holistic overview of any practice

or service that is related to racism, but not explicitly framed or analyzed from an (anti-)racism perspective. For example, we did not capture more of the literature documenting the role of PHA in the Holocaust, since they either predominantly used terms like hereditary/racial welfare and Nazism (e.g., [103–106]). Since we did not use antisemitism as a search term, the findings might reflect on the contested relationship of antisemitism and racism, which are criticized to be constructed as exclusionary concepts [4]. Yuval-Davis is therefore calling for an inclusive definition of antisemitism as a form of racism [4]. Also, work focused on culturally-sensitive discourse was outside the scope of our review (e.g., [101]), even though they might complement approaches to dismantle racism in PHA. We acknowledge this as a site of silence in our review.

Another limiting factor is the definition of public health services, which is highly context-specific and not clear-cut. It is often the case that PHA are entangled with care provision, which resulted in a high number of records screened at the full-text stage. Since we were not able to assess this differentiation based on the abstracts, a lot of the studies and articles were excluded after the full-text screening. Furthermore, some of the barriers impeding access to public health services, e.g., language barriers, realities of living (housing, infrastructure), knowledge production processes [107], disproportionate testing patterns could also be interpreted as a manifestation of institutional racism but were often not framed or conceptualized as such. Thus, we did not include these since they do not directly contribute to the scientific discourse on racism in PHA but acknowledge their indirect influence, e.g., [107].

### Implications for practices in public health services and PHA

Our findings highlight the imperative for PHA to adopt anti-racist practices that actively dismantle systemic and institutional biases embedded in their frameworks and services. A critical step is breaking down silos, as PHA often work closely with other public services, such as child welfare systems, law enforcement, and the legal justice system. These connections emphasize the interconnectedness of public health with broader governmental structures and the responsibility of PHA to challenge and transform racialized practices and policies across all levels of governance and service delivery.

Addressing epistemic violence - that is giving voice to the (historical as well as present) sites of silences - is essential in reshaping how PHA operate and services are organized [31]. PHA and associated research need to critically evaluate which kinds of seemingly neutral categories data collection in health monitoring is (re-)producing, and if categories like 'migrant' are representative of the homogenous needs within highly diverse migrant populations [108]. In other words, PHA as well as public health scholars need to critically reflect on how knowledge is produced, who is included in decision-making processes, and how marginalized voices can be meaningfully centered. Efforts should be directed at replacing exclusionary practices with inclusive, equity-driven frameworks that prioritize the lived experiences of affected communities. This transition requires moving away from top-down approaches and fostering collaborative partnerships with racialized communities to co-create policies that address systemic racism and promote health equity.

As public institutions, PHA have a responsibility to critically reflect on their past and consider reparative measures for historical and ongoing injustices. For this, public health scholars should work closely together with PHA to share knowledge resources and positive examples like those cited in this literature review, to encourage more critical, and sometimes uncomfortable, questions around racism. The enduring legacy of colonial structures and historical events remains a relevant reference point for racialized groups, shaping their trust in and engagement with public health systems. Anti-racist practices must confront these colonial legacies by integrating restorative approaches that build trust and address historical wrongs [5]. This includes acknowledging their role in perpetuating racial inequities and actively working towards repairing and preventing these harms.

### Supporting information

**S1 Checklist. Preferred Reporting Items for Systematic reviews and Meta-Analyses extension for Scoping Reviews (PRISMA-ScR) Checklist, licensed under a CCBY 4.0 license.**
(DOCX)

## S1 Text. Search protocols.
(DOCX)

## S1 Table. Data charting of included sources/ actants.
(DOCX)

## Acknowledgments

We honor the memory of Fadeke Berida, whose invaluable contributions and unwavering commitment greatly enriched this research. We thank Prof. Dr. Dagmar Starke, Academy of Public Health Services, who encouraged us strongly to conduct this review and supported us throughout with enlightening discussions.

## Author contributions

**Conceptualization:** Oliver Razum, Yudit Namer.

**Formal analysis:** Theresa Altmiks, Julia Zielke.

**Funding acquisition:** Oliver Razum, Yudit Namer.

**Investigation:** Lisa Wandschneider, Sigsten Stieglitz, Theresa Altmiks, Yudit Namer.

**Methodology:** Lisa Wandschneider, Julia Zielke, Yudit Namer.

**Writing – original draft:** Lisa Wandschneider, Sigsten Stieglitz, Oliver Razum, Julia Zielke, Yudit Namer.

**Writing – review & editing:** Lisa Wandschneider, Sigsten Stieglitz, Theresa Altmiks, Oliver Razum, Julia Zielke, Yudit Namer.

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
