## [Decision Letter · Decision Letter 0]

26 Apr 2025

PGPH-D-25-00327

Racism in public health authorities – a scoping review and situational analysis

Dear Dr. Razum,

Thank you for submitting your manuscript to PLOS Global Public Health. After careful consideration, we feel that it has merit but does not fully meet PLOS Global Public Health’s publication criteria as it currently stands. Therefore, we invite you to submit a revised version of the manuscript that addresses the points raised during the review process.

Both reviewers have indicated the substantial value of your review and provided focal comments to be addressed in the Introduction, Methods, and Discussion sections, which would be beneficial to the final manuscript. In addition to Reviewer 1's helpful requests for clarifications, Reviewer 2 makes several important constructive suggestions throughout the manuscript, all of which appear to be relatively straightforward to address. 

We look forward to receiving your revised manuscript.

Kind regards,

Peter A. Newman, Ph.D

Academic Editor

Journal Requirements:

Additional Editor Comments (if provided):

Reviewers' comments:

Reviewer's Responses to Questions

**Comments to the Author**

1. Does this manuscript meet PLOS Global Public Health’s publication criteria?

Reviewer #1: Yes

Reviewer #2: Yes

2. Has the statistical analysis been performed appropriately and rigorously?

Reviewer #1: N/A

Reviewer #2: Yes

3. Have the authors made all data underlying the findings in their manuscript fully available (please refer to the Data Availability Statement at the start of the manuscript PDF file)?

Reviewer #1: Yes

Reviewer #2: Yes

4. Is the manuscript presented in an intelligible fashion and written in standard English?

Reviewer #1: Yes

Reviewer #2: Yes

Reviewer #1: Thank you for this interesting and important scoping review. It is well-written, with excellent structure and a strong methodology. The findings have important implications for health authorities.

Was grey literature searched in this review, e.g. reports, blogs? If not, why not? A sentence justifying the exclusion of this form of knowledge from the scoping review (especially given how racialisation shapes whose knowledge counts and access to academic knowledge production) would be important to add to the methods.

In a paper on a topic like this, it is important for the authors to reflect on their positionality in the methods section, especially their racial and ethic backgrounds and how this shaped the focus on this topic.

Reviewer #2: I appreciated the opportunity to review your manuscript. I have offered comments in the spirit of strengthening your paper. The scoping review addresses an important and timely topic of racism in public health agencies. It is overall well-written.

Abstract:

- could benefit from a brief definition or at least mention of racism as a structural determinant of health (as you have done in the introduction to the paper).

- "have been neglected as relevant contexts" - Change to institutional contexts. The use of contexts on its own is vague. consider different wording.

- What scoping review methodology was used? Suggest that you name briefly in the abstract

Introduction

- strong overview of racism as a structural determinant of health and PHAs.

- Can you elaborate a bit more on how you arrived at the gap involving PHAs (is this simply a gap in the peer reviewed literature that has received little to no attention? Is this because less is published on PHAs relative to health care organizations where funding for health (or illness) is disproportionately directed?). Given PHAs' mandate, they should conceivably be more concerned with equity, social justice and racism.

Research questions are clear - did the authors predominantly engage with structural forms of racism (i.e. institutional/systemic) or also interpersonal and individual as well? (from the discussion section, it appears that you were quite inclusive)

Methods - overall this section is clearly detailed. I appreciated the application of both a narrative synthesis and situational analysis

Briefly, what approach to a scoping review did you use? needs to be more explicit. Here are two commonly used references: https://www.tandfonline.com/doi/abs/10.1080/1364557032000119616 and https://link.springer.com/article/10.1186/1748-5908-5-69

search strategy - the search strategy is comprehensive. It's unclear what the publication dates were used to delimit the search (articles included were published between X date and Y date and why this timeframe was selected)

Given that some organizations may offer both preventive and curative services, how did you contend with such cases? Were such articles excluded?

Results & discussion

in the opening line to this section, it would be helpful to briefly mention how many were originally considered. Of the XXX that were identified through our search strategy, we included XX after XXX screening, and then 55 publications after full text screening.

Are you able to summarize whether the focus of PHA featured in the articles included was at the national, sub-national (e.g. local or regional) or international level?

Conceptualization of racism - for the papers that did define racism, did any reference social theory (e.g. critical race theory) in their analyses? If yes, which theories or frameworks? If not, this may be worth mentioning as an analytic point in the discussion and future work

I appreciated your analysis of how PHAs socially constructed high risk groups resulting in othering and also homogenizing population groups (e.g. Latinx). Another term that has received considerable attention is 'vulnerable'. It showed up in your results as well (the critique is that it puts the blame on the populations rather than drawing attention to the structures that make populations vulnerable - some have shifted the use of language away from vulnerable to populations made structurally vulnerable. Did this emerge in your analysis as well?

In presenting your results, you implicitly refer to different functions of PHAs (e.g. surveillance, programs, policies, guidelines, legal authority/oversight). What were the gaps in terms of PHA functions. For example, health promotion (which is part of the remit of PHA) wasn't mentioned but likely a limitation of what was in your included sample of articles

Table 2 is excellent - here you do give some useful examples of PHAs operating at different levels (see comment above). I appreciate the challenges given the limited inclusion of potentiality statements. This may point to the likely omission of using theory/theoretical frameworks (see earlier comment)

Strengths and Limitations are well outlined. Is it possible that there may be a publication bias (less written on PHAs relative to health care orgs), recognizing the challenges with the definition of public health services.

Implications section focuses on practice - are there also any implications for future research that your findings should point to?

Minor points

line 236 - roles should be plural

line 425 - the findings might reflect on the on the contested relationship - delete 'on the' which appears twice -

**Do you want your identity to be public for this peer review?** For information about this choice, including consent withdrawal, please see our Privacy Policy

Reviewer #1: No

Reviewer #2: No

---

## [Editor Report · Decision Letter 1]

6 Nov 2025

Racism in public health authorities – a scoping review and situational analysis

PGPH-D-25-00327R1

Dear Prof Razum,

We are pleased to inform you that your manuscript 'Racism in public health authorities – a scoping review and situational analysis' has been provisionally accepted for publication in PLOS Global Public Health.

Best regards,

Julia Robinson

Executive Editor